# Small-Scale Randomized Controlled Trial to Explore the Impact of β-Hydroxy-β-Methylbutyrate Plus Vitamin D_3_ on Skeletal Muscle Health in Middle Aged Women

**DOI:** 10.3390/nu14214674

**Published:** 2022-11-04

**Authors:** William D. Fairfield, Dennis M. Minton, Christian J. Elliehausen, Alexander D. Nichol, Taylor L. Cook, John A. Rathmacher, Lisa M. Pitchford, Scott A. Paluska, Adam J. Kuchnia, Jacob M. Allen, Adam R. Konopka

**Affiliations:** 1Department of Kinesiology and Community Health, University of Illinois Urbana-Champaign, Urbana, IL 61801, USA; 2Division of Geriatrics and Gerontology, Department of Medicine, University of Wisconsin-Madison, Madison, WI 53705, USA; 3Geriatrics Research Education and Clinical Center (GRECC), William S. Middleton Memorial Veterans Hospital, Madison, WI 53705, USA; 4Department of Nutritional Sciences, University of Wisconsin-Madison, Madison, WI 53706, USA; 5MTI BioTech, Inc., Ames, IA 50010, USA

**Keywords:** intermuscular adipose tissue (IMAT), sarcopenia, hypertrophy, resistance exercise

## Abstract

*β*-Hydroxy-*β*-methylbutyrate (HMB), a leucine metabolite, can increase skeletal muscle size and function. However, HMB may be less effective at improving muscle function in people with insufficient Vitamin D_3_ (25-OH-D < 30 ng/mL) which is common in middle-aged and older adults. Therefore, we tested the hypothesis that combining HMB plus Vitamin D_3_ (HMB + D) supplementation would improve skeletal muscle size, composition, and function in middle-aged women. In a double-blinded fashion, women (53 ± 1 yrs, 26 ± 1 kg/m^2^, *n* = 43) were randomized to take placebo or HMB + D (3 g Calcium HMB + 2000 IU D per day) during 12 weeks of sedentary behavior (SED) or resistance exercise training (RET). On average, participants entered the study Vitamin D_3_ insufficient while HMB + D increased 25-OH-D to sufficient levels after 8 and 12 weeks. In SED, HMB + D prevented the loss of arm lean mass observed with placebo. HMB + D increased muscle volume and decreased intermuscular adipose tissue (IMAT) volume in the thigh compared to placebo but did not change muscle function. In RET, 12-weeks of HMB + D decreased IMAT compared to placebo but did not influence the increase in skeletal muscle volume or function. In summary, HMB + D decreased IMAT independent of exercise status and may prevent the loss or increase muscle size in a small cohort of sedentary middle-aged women. These results lend support to conduct a longer duration study with greater sample size to determine the validity of the observed positive effects of HMB + D on IMAT and skeletal muscle in a small cohort of middle-aged women.

## 1. Introduction

Age-related loss of skeletal muscle mass begins during the fourth decade of life and is a leading risk factor for the development of chronic disease and disability. Sedentary lifestyle combined with aging also changes the composition of skeletal muscle as evident by the accumulation of intermuscular adipose tissue (IMAT). IMAT is the fatty infiltration in skeletal muscle beneath the muscle fascia, which includes ectopic lipid deposition both in between muscle groups and in between muscle fibers. Elevated IMAT is a predictor of worsened muscle strength and physical function and leads to future risk of mobility limitations in older adults [1,2,3,4]. Excess IMAT infiltration is also linked to attenuated hypertrophic response to resistance exercise training [5].

Throughout adulthood, women, on average, have lower skeletal muscle size and function than their male counterparts. Older women have greater infiltration of IMAT compared to younger and older men and greater IMAT infiltration was inversely correlated with skeletal muscle function and exercise capacity [6]. Menopause significantly impacts body composition, accelerating the decline in skeletal muscle mass with a parallel increase in adipose tissue [7]. Collectively, changes in muscle size, function and composition are associated with a greater risk of multimorbidity and disability in women [8,9]. While women currently have longer median lifespan than men, the additional years may be vulnerable to functional limitations and disability [10]. Therefore, identifying strategies to combat the age-related loss of muscle size, function, composition and quality are critical to extend healthy lifespan, especially in women.

While exercise is the gold-standard approach to elicit skeletal muscle hypertrophy and enhance function [6,11,12,13], there is significant heterogeneity with some individuals failing to show improvements [14]. Therefore, supplementary non-pharmacological strategies to augment skeletal muscle health in combination with exercise are needed. Further, identifying alternative approaches to improve skeletal muscle health could also be advantageous for those who cannot or do not meet physical activity recommendations. The leucine metabolite *β*-Hydroxy-*β*-methylbutyrate (HMB) can stimulate muscle protein synthesis and decrease muscle protein breakdown in humans [15,16]. These findings build on pre-clinical work that suggest HMB stimulates muscle protein synthesis while attenuating proteolysis during inflammatory conditions common to aging [17,18]. The anti-catabolic effects of HMB can help minimize the loss of muscle mass during unloading [19,20] and cancer cachexia [21]. In non-catabolic conditions, HMB with or without exercise can have positive benefits on muscle mass and function in both young and older adults [22,23] although this is not universal [24]. HMB was shown to have an equivalent effect compared to leucine on skeletal muscle hypertrophy after resistance exercise training in younger, trained men [25]. The positive effects of HMB on skeletal muscle mass or function are largely consistent with previous studies that have used HMB in combination with other amino acids to show modest increases in muscle mass and some measures of function [26,27,28]. HMB has also been shown to aid in fat loss when taken with resistance exercise training [29,30]. Collectively, these data suggest that HMB could improve net protein balance conducive for maintenance of muscle and promote fat loss.

Vitamin D_3_ insufficiency and deficiency are often associated with lower muscle mass and function with aging. Older (65–88 yrs) and middle-aged (55–65 yrs) adults with Vitamin D_3_ insufficiency presented with more functional limitations than those with Vitamin D_3_ sufficiency (>30 ng/mL) [31]. Further, Vitamin D_3_ insufficiency was associated with an increase in functional limitations after 3 and 6 year follow up in the older and middle-aged group, respectively [31]. Even starting at younger age, Vitamin D_3_ insufficiency is accompanied with greater IMAT infiltration in women than those who are vitamin D_3_ sufficient [32]. These data suggest that Vitamin D_3_ may be important in regulating skeletal muscle size, composition, and function across the lifespan.

Intriguingly, a retrospective analysis revealed that only individuals who were Vitamin D_3_ sufficient increased muscle function after supplementation with a cocktail of HMB, arginine, and lysine, while individuals who were vitamin D_3_ insufficient did not [33]. In line with these findings, the anabolic effects of leucine plus insulin on C2C12 myotubes are greater with Vitamin D_3_ supplementation [34]. These data may indicate that to maximize the anabolic potential of leucine or leucine metabolites like HMB, there may be a need for sufficient Vitamin D_3_ levels.

Recently, co-supplementation of HMB plus Vitamin D_3_ (HMB + D) increased lean mass and composite measure of physical function in older adults [35]. Further, HMB + D prevented the decline in leg extension peak torque over 12 months of sedentary behavior [35]. However, it remains unknown if the benefits of HMB + D in older adults can be extended to middle-age during the early stages of age-related loss of skeletal muscle size and function. Therefore, middle-age is a critical time to intervene to help delay or slow the loss of skeletal muscle health and mitigate the risk for future limitations in mobility and activities of daily living. The purpose of this study was to test the hypothesis that supplementation with calcium HMB + D, aimed at achieving sufficient circulating levels of 25-OH-D, could improve skeletal size, function, composition, and quality in middle-aged women (45–60 yrs) during 12-weeks of a non-exercise sedentary control (SED) or resistance exercise training (RET) program.

## 2. Methods

### 2.1. Subjects

This study was approved by the Institutional Review Board at University of Illinois and registered as a clinical trial (NCT03848741), and performed at the University of Illinois Urbana-Champaign. Data analysis was completed at the University of Wisconsin-Madison. All interested participants provided written, informed consent after detailed review of the study procedures, risks and potential benefits. We enrolled and started data collection for our target sample size of 48 women (*n* = 12 per group) but 10 women were impacted by study shutdown during COVID-19. None of these women chose to return to the study after re-opening. We recruited 5 additional subjects comfortable with completing the study during the unpredictable and unknown impact of COVID-19. Recruitment was conducted before many individuals had access to vaccines. Therefore, 43 women completed this small-scale study. Each participant completed a medical and health history questionnaire, an international physical activity questionnaire (IPAQ), a 36-item Short Form questionnaire and a blood chemistry profile. Participants were excluded based on the following criteria (1) body mass index ≥35 kg/m^2^; (2) type 1 or type 2 diabetes; (3) uncontrolled hypertension; (4) active cancer, cancer in remission, or having received treatment for any form of cancer in the previous 5 yrs; (5) cardiovascular disease (e.g., peripheral arterial disease, peripheral vascular disease); (6) uncontrolled thyroid function; (7) engaged in regular resistance exercise more than two times per week for 20 min or longer during the previous year; (8) regular nonsteroidal anti-inflammatory drug consumption; and (9) any condition that presents a limitation to exercise training (e.g., severe arthritis, chronic obstructive pulmonary disease, neuromuscular disorder, cognitive impairment, Alzheimer’s disease, vertigo, dizziness).

### 2.2. Overview of Study Design

Eligible volunteers underwent a series of baseline measurements for serum 25-OH-D and complete blood panel, urine HMB, determination of skeletal muscle function via one- and ten-repetition maximum (1RM, 10RM) and maximal isokinetic knee extension, body composition assessment via dual energy X-ray absorptiometry (DEXA), and assessment of thigh muscle and IMAT volume with magnetic resonance imaging (MRI). Participants also completed a 3-day dietary log to estimate macro- and micronutrient dietary composition and questionnaires to estimate physical activity and quality of life. Blood draws, urine collection, and 1RM/10RM (for exercise groups only) were repeated after 4, 8, and 12 weeks. Following the intervention, all subjects repeated all baseline testing procedures. Unblinding occurred after data analysis was complete.

### 2.3. Randomization and Study Blind

Subjects were randomized to one of four groups which included placebo or HMB + D supplementation during 12 weeks of non-exercise sedentary control (SED) or a progressive resistance exercise training program (RET). The randomization sequence was created in STATA and was performed in random block sizes of 4 or 8 to minimize selection bias. The randomization was performed by the investigative team at UIUC/UW-Madison and was concealed to all study team members, including those employed by MTI. Further, the study blind was maintained by the study team until after data analysis.

### 2.4. Dietary Assessment

Participants were provided with written and oral instructions to record their dietary intake over 3 days, including portion sizes, content, and types of food. After reviewing the food diary with participants, the research team analyzed the data via ESHA Food Processor Nutrition Analysis software to estimate macro and micronutrient intake at weeks 0 and 12.

### 2.5. Body Composition and Sample Collection

After an overnight fast, participants arrived for body composition, blood and urine collection. Whole body and regional mass, fat mass, and lean mass were assessed by whole body DEXA scans (Hologic QDR 4500A). Blood was collected in an EDTA vacutainer and serum separator tube for the measurement of a complete blood chemistry panel and 25-OH-D concentrations. Urine was aliquoted into urinalysis tube for urine HMB concentrations and urine analysis. At 12 weeks, blood and urine samples were collected 36 h after the last exercise session and/or capsule. LabCorp collected and analyzed all blood chemistry panels, and Heartland Assays (Ames, IA) analyzed blood 25-OH-D via the Liaison XL automated chemiluminescence analyzer and urine HMB by GCMS [35].

### 2.6. Magnetic Resonance Imaging

Proton MR images of the thigh were evaluated before and after the 12-wk intervention using a Prisma 3.0 Tesla imaging system at standard settings (TR/TE = 2000/9 ms), similar to previously protocols [6,11,12,36,37,38]. Scans at 12 weeks were obtained 36–48 h after an exercise bout. All MRI scans were performed after a minimum of 15 min of supine rest to minimize the influence of potential fluid shifts [11]. Subjects were positioned with an adjustable foot restraint to control for joint angles and thus, muscle lengths before and after the intervention as previously performed [11]. A 64-channel coil was placed around the thigh. Contiguous, 1-cm interleaved serial scans were obtained from the head of femur to 10 cm below the tibial plateau.

Prior to analysis, post- scans were first co-registered with pre-scans using SPM12 to account for subtle differences in thigh position that could not be controlled for during image acquisition. Scans were then cropped to contain a volume of interest ranging from the first distal image containing musculus rectus femoris and the last proximal image not containing the gluteal muscle, and a bias field correction was applied with gaussian smoothness ranging from 60–40 mm depending on the severity of signal dropout. Default settings were used for all other bias field correction parameters. As shown in Figure 1, image stacks were imported into Amira 6.0 (Thermo) and segmented to create a binary image stack containing fatty tissue (subcutaneous fat and IMAT). Morphological operations (object closing/filling and image subtraction) were then performed to isolate subcutaneous adipose, IMAT, and skeletal muscle in the thigh. Tissue volumes were then determined by multiplying the voxel counts for subcutaneous fat, IMAT, and muscle by voxel size acquired during scanning. After scanning, images for one participant in the SED HMB + D group were not saved and could not be used for downstream analysis.

### 2.7. Muscle Function Testing

All participants completed a 5-min warm up on a stationary bicycle before any functional assessments. During a visit prior to testing, all subjects were fitted to each piece of equipment (Life Fitness) and completed a familiarization session where they practiced full range of motion using proper technique with increasing, non-maximal workloads. 1-RM were assessed for leg press, leg extension, and leg curl exercises. 10-RM were assessed for chest press, shoulder press, and seated row exercises. Testing was divided into two days separated by a minimum of 72 h. The first testing day consisted of dynamometry testing, leg curl, chest press, and seated row. The second testing day consisted of leg press, leg extension, and shoulder press.

Before and after the 12-week intervention, participants performed a maximal isokinetic knee extension contractions at three different velocities (60, 120, 180° per second) on a biodex dynamometer (Biodex System 3).

### 2.8. Resistance Exercise Training Program or Sedentary Control

Whole-body progressive resistance exercise training was performed 3 times/week for 12 weeks under supervision by an exercise physiologist using a similar protocol as previously described [13]. Upon arrival for each exercise bout, participant body weight was recorded. Participants completed a 5-min warm up on a cycle ergometer prior to each training session. Each training session alternated between (1) leg press, leg extension, chest press, and shoulder press and (2) leg press, leg curl, seated row, bicep curl, and tricep extension. Participants started with 2 sets of 10–15 repetitions the first week, 3 sets the second week and 4 sets from weeks 3 to 12. Training intensity was based on 70–80% of 1RM for lower body exercises and 80–90% 10RM for upper body exercises. During the first 2 weeks of training, the training intensity was increased from 70% 1RM (10–15 repetitions) to 80% 1RM (10–12 repetitions). The 1RM and 10RM were re-evaluated every 4 weeks and a new 70–80% 1RM and 80–90% 10RM were calculated. Weight was increased for bicep curl and triceps extension when 14 repetitions could be completed while still maintaining proper technique.

Participants randomized to the non-exercise sedentary control group were instructed to not change their physical activity during the trial and were offered 8 weeks of supervised exercise training after completion of the study.

### 2.9. Supplementation

Subjects in the SED and RET were randomized, in a double-blind fashion to placebo (calcium lactate) or HMB + D (3 g of calcium HMB plus 2000 IU of Vitamin D_3_). 1.5 g of HMB and 1000 IU of Vitamin D_3_ were taken twice daily. Subjects consumed study capsules with meals or snacks. On exercise days, subjects completing resistance exercise training consumed study capsules ~30–45 min prior to their training session. The placebo group consumed capsules containing calcium lactate which were equal in size, color, weight, and capsule number per dose. Supplements were supplied by Metabolic Technologies LLC (Ames, IA) and manufactured by TSI group, Ltd. (Missoula, MT). Placebo and HMB + D capsules contained the same amount of calcium, phosphorous, and potassium. All participants were given a random number of additional pills and compliance was estimated based on the number of pills returned. Pills were exchanged every 2 weeks.

### 2.10. Statistical Analysis

The primary endpoint was thigh skeletal muscle volume as assessed by MRI. Since no studies using HMB + D have been previously performed in middle-aged women, power analysis (G-Power, Universität Kiel, Germany) was completed based on the change in lean body mass by Rathmacher et al. [35] testing HMB + D in older adults. For the power analysis, we used a previously measured 0.44 kg change in lean body mass with HMB + D and a −0.37 Kg change in lean body mass for placebo to support our primary outcome of thigh skeletal muscle volume. Based on a F-Test (ANOVA) with repeated measure for time (pre vs. post) and 4 groups with a power of 0.8 and an α-error probability of 0.05, we estimate the need for a total of 44 subjects.

All data are presented as mean ± standard error (SE) for pre vs. post measures as well as the individual data for the change from pre to post. Since the goal of this study was to determine the impact of HMB + D in middle-aged women randomized to sedentary or resistance exercise training groups, all statistical analyses were completed within each group (SED and RET) and following a per-protocol analysis. An a priori compliance rate was set at 80%, which permitted subjects to miss up to 1 day of capsules per week and was lower than our average compliance rates in previous studies using nutritional supplementation or commonly consumed medications (≥90%) [35,39,40,41]. Statistical significance was determined with a two-way (treatment × time) ANOVA with repeated measures for time. In the presence of an interaction, Holm-Sidak’s post hoc test was performed. The absolute change from pre to post was also compared using an unpaired *t*-test (HMB + D vs. placebo). Statistics were performed using Prism Version 9 (GraphPad, San Diego, CA). Significance was set a priori at *p ≤* 0.05.

## 3. Results

### 3.1. Subject Characteristics

A CONSORT study flow diagram is presented in Appendix A. A total of 4 participants were non-compliant with the capsule protocol (<80% of study capsules) and were removed from the per protocol analysis. Of the non-compliant participants, 1 was in the SED HMB + D, 2 in the RET placebo and 1 in the RET HMB + D. Otherwise, study capsule compliance was 94% for placebo and 92% for HMB + D. Exercise compliance was 100%. All data presented are from the per protocol analysis.

Serum 25-OH-D and urinary HMB levels at baseline, 4, 8 and 12 weeks are shown in Figure 2 and blood chemistry, lipids, and hematology are shown in Appendix A. At baseline, both groups were, on average, Vitamin D_3_ insufficient (<30 ng/mL). HMB + D increased circulating total 25-OH-D to sufficient levels after 8 (*p* < 0.05) and 12 (*p* < 0.01) weeks of supplementation. HMB + D increased urinary HMB after 4 and 8 weeks but no differences were observed at 12 weeks. Urine samples were taken 36 h after the last capsule and therefore HMB likely returned near baseline at 12 weeks. There was no significant effect of placebo on 25OH-D or HMB levels.

Self-reported dietary intake is shown in Table 1. Caloric and macronutrient intake was not different before versus after the intervention for any group (weeks 0 vs. 12).

There were no changes in SF-36 scores for SED. RET increased SF36 scores (*p* < 0.05, main effect for time) for general health, energy, and emotional well-being but there was no difference between placebo versus HMB + D.

### 3.2. Whole Body and Regional Tissue Mass

Table 2 shows DEXA derived whole body and regional lean and fat mass. At baseline, there were no statistical differences between groups. In SED, there was no significant effect of HMB + D on whole body lean mass, sarcopenic index, or leg lean mass. However, there was a significant treatment x time interaction (*p* < 0.01) for arm lean mass where the control group decreased (*p* < 0.05) while the HMB-D maintained arm lean mass. HMB + D did not influence whole body, trunk, leg, or arm fat mass.

Resistance exercise training increased (*p* < 0.001, main effect for time) whole body, leg, and arm lean mass but there was no difference between HMB + D versus placebo. Similarly, resistance exercise training increased (*p* < 0.001, main effect for time) the sarcopenic index but there was no difference between HMB + D versus placebo. Whole body and regional fat mass were not significantly influenced by HMB + D or resistance exercise training.

### 3.3. Skeletal Muscle Size and IMAT

In Figure 3A–D, group means are provided on the left and individual changes on the right. In the SED, there was a treatment × time interaction (*p* = 0.05) for thigh skeletal muscle (Figure 3A) and IMAT volume (Figure 3B). The increase in thigh muscle volume (26 ± 13 cm^3^ vs. −5 ± 8 cm^3^) and the decrease in IMAT (−20 ± 12 cm^3^ vs. 6 ± 6 cm^3^) was greater (*p* = 0.05) for HMB + D versus placebo. RET increased (*p* < 0.001, main effect of time) thigh muscle volume but there was no influence of HMB + D on thigh muscle hypertrophy versus placebo (+107 ± 20 cm^3^ vs. +111 ± 26 cm^3^; Figure 3C). In RET, there was a treatment x time interaction (*p* = 0.05) for IMAT. HMB + D decreased (*p* = 0.05) IMAT versus placebo (−22 ± 12 cm^3^ vs. 6 ± 4 cm^3^; Figure 3D).

### 3.4. Muscle Function

Figure 4 shows leg muscle function. In the SED, there were no differences in 1RM leg extension or flexion (Figure 4A,B). There was an effect of time in both HMB + D and placebo for increased 1RM leg press, suggesting a learning effect despite the use of familiarization protocols (Figure 4C). There was no influence of HMB + D on isokinetic assessments of peak torque production at three different velocities (60, 120, 180°/s; Figure 4D).

In RET (Figure 4E–H), the increase (*p* < 0.01, main effect for time) in leg extension, flexion, and press was not different between HMB + D versus placebo. Similarly, the increase (*p* < 0.05, main effect for time) in peak isokinetic torque production at 60, 120, and 180°/s after resistance exercise training was not different between HMB + D versus placebo.

Table 3 shows upper body muscle function. In SED, there was a main effect for time for 10RM chest press that was driven by a within group increase (*p* < 0.05) in the HMB + D group. There were no differences in shoulder press or seated row. In the RET, the increase (*p* < 0.01, main effect for time) in chest press, shoulder press, and seated row after resistance exercise training was not different between HMB + D versus placebo.

### 3.5. Muscle Quality

In Figure 5, muscle strength is expressed relative to thigh muscle volume to determine specific muscle function or muscle quality. In SED, there was a treatment × time interaction, where the HMB + D group decreased leg extension specific force (Figure 5A). There were no differences in specific force for leg press or flexion (Figure 5B,C). Resistance exercise training increased specific force (*p* < 0.01, main effect of time) but there were no differences between HMB + D versus control for leg extension, leg press or leg flexion (Figure 5D–F).

### 3.6. Exploratory Analysis: Role of Vitamin D_3_ Sufficiency on the Impact of HMB on IMAT

While on average HMB + D increased 25-OHD from insufficient to sufficient levels in both the SED and RET, there was noticeable heterogeneity and some individuals remained Vitamin D_3_ insufficient. Since IMAT was the only variable that decreased (~20 cm^3^) consistently in both SED and RET receiving HMB + D, the data were collapsed to explore the potential impact of Vitamin D_3_ on IMAT volume. In the HMB + D supplementation groups we compared individuals who finished the study Vitamin D_3_ insufficient (*n* = 7) vs. sufficient (*n* = 11). As shown in Figure 6A, in the HMB + D supplementation groups there was a within-group decrease in IMAT in those who finished the study Vitamin D_3_ sufficient but no change in those who finished the study Vitamin D_3_ insufficient. Further, since some individuals started the study Vitamin D_3_ sufficient while others became Vitamin D_3_ sufficient after HMB + D supplementation, individuals were further stratified into those who remained insufficient, remained sufficient or transitioned from insufficient to sufficient after HMB + D (Figure 6B). These exploratory analyses reveal potential trends where individuals who went from Vitamin D_3_ insufficient to sufficient may have the greatest decrease in IMAT after HMB + D supplementation, albeit these findings were not statistically significant and included subjects across both SED and RET groups.

## 4. Discussion

The major findings of this study suggest that 12-weeks of HMB + D supplementation decreased thigh IMAT volume in middle-aged women who remained sedentary or completed resistance exercise training. Further, HMB + D supplementation decreased IMAT in women who completed the study as Vitamin D_3_ sufficient but not those who remained Vitamin D_3_ insufficient. In non-exercising sedentary middle-aged women, HMB + D maintained arm lean mass and increased thigh skeletal muscle volume compared to placebo. However, the changes in thigh composition after 12 weeks of HMB + D supplementation did not improve muscle function. While resistance exercise training increased thigh skeletal muscle volume, function, and muscle quality, there was no influence of HMB + D. Therefore, HMB + D appears to maintain muscle size in sedentary conditions and decrease IMAT in middle aged women with Vitamin D_3_ sufficiency who remain sedentary or complete resistance exercise training. Additional follow-up studies are warranted to determine if these findings can be validated in a larger sample size and if HMB + D can improve health outcomes in populations with high levels of IMAT infiltration.

Using gold-standard MRI, we demonstrate differences in skeletal muscle volume between 12 weeks of HMB + D supplementation versus placebo in the SED group, indicating that HMB + D may have positive effects on skeletal muscle maintenance or growth. However, HMB + D did not augment skeletal muscle hypertrophy after resistance exercise training. These findings are in agreement with a previous study showing that HMB + D increased whole-body lean mass in older adults after 6 but not 12 months of supplementation [35]. In the current study, the amount of hypertrophy observed with HMB + D supplementation alone was approximately 25% of the hypertrophy observed with resistance exercise training. Therefore, for middle-aged women who are unable or unwilling to perform resistance exercise training, HMB + D supplementation may be beneficial to maintain or induce a subtle increase in muscle volume. In further support of this notion, our results show that the SED placebo group decreased lean arm mass while HMB + D supplementation did not. These results are consistent with previous work demonstrating the anti-catabolic effects of HMB during bedrest [19] and unloading [20] and suggest HMB + D supplementation may have spared muscle loss specific to the arms during 12-weeks of sedentary behavior. We propose that arms remain relatively unused during normal ambulation in sedentary participants compared to the leg muscles. Overall, this study provides initial data to suggest that HMB + D supplementation may be beneficial in middle aged women to maintain or prevent the loss of muscle mass.

To our knowledge, this is the first study to investigate if HMB + D influences the infiltration of IMAT in middle-aged women. We show differences between HMB + D versus placebo to suggest that 12 weeks of HMB + D supplementation may decrease IMAT, regardless of exercise training. Further, since body weight and dietary intake were not statistically different, these data further support the link between the decrease in IMAT and HMB + D. These findings in middle-age women parallel previous data in pre-frail older adults where 12-weeks of HMB decreased IMAT [27]. However, in the current study, HMB + D supplementation seemed to decreased IMAT infiltration only in women who were Vitamin D_3_ sufficient or became sufficient after supplementation. Moreover, women who transitioned from Vitamin D_3_ insufficient to sufficient appeared to have the greatest decrease in IMAT after HMB + D, albeit these data were limited by sample size and not statistically significant. The association between HMB + D and IMAT coincide with other work that indicate inverse relationships between vitamin D_3_ levels and IMAT [32]. While the mechanisms by which HMB + D lowers IMAT remain unknown, there is evidence to indicate that HMB and Vitamin D_3_ can independently increase or maintain skeletal muscle mitochondrial biogenesis and/or respiratory capacity in vitro and in rodents [42,43,44]. Additional work in humans is needed to identify whether changes in mitochondrial metabolism by HMB + D is related to the change in IMAT and/or skeletal muscle volume. Collectively, these data suggest that sufficient Vitamin D_3_ levels could contribute to decreased IMAT after HMB supplementation but this conjecture needs to be directly tested.

Previous studies have demonstrated that baseline IMAT and the change in IMAT were correlated to the increase in skeletal muscle size and function after exercise training programs [5,6]. Therefore, we reasoned that since HMB + D supplementation promoted the loss of IMAT, HMB + D would also augment improvements in skeletal muscle growth and function after resistance exercise training, however, this was not observed. IMAT has also been correlated to skeletal muscle insulin sensitivity [45,46] and therefore, future studies should also consider evaluating if a decrease in IMAT by HMB + D is accompanied by improvements in insulin sensitivity and indices of metabolic health. Patients recovering from disuse after injury, orthopedic surgery, and those with cancer have increased IMAT infiltration which is associated with impaired function, rehabilitation, and survival [47,48,49,50]. Therefore, these data suggest the need to test if HMB + D could be a valuable non-pharmacological tool to (1) prevent the accumulation or (2) promote the loss of fatty infiltration during periods of impaired metabolic health, immobility or rehabilitation. If successful, these studies could identify additional populations that could potentially benefit from HMB + D when moderate to vigorous intensity exercise training may not be feasible.

Despite HMB + D supplementation decreasing IMAT in both sedentary and exercising women and modestly increasing thigh muscle volume in sedentary women, the improvements in thigh composition did not translate to improvements in function. A retrospective analysis identified an association where those who were Vitamin D_3_ insufficient did not improve muscle function after an amino acid cocktail containing HMB while those who were Vitamin D_3_ sufficient increased muscle function [33]. Adding Vitamin D_3_ to 12 months of HMB supplementation in older adults increased a composite score of physical function and prevented the loss of peak knee extensor torque [35]. A key difference between the current study and previous work in older adults include investigating the influence of HMB + D on skeletal muscle function in middle-aged women where the decline in function may be occurring at a slower rate or a lesser magnitude as compared to older adults. Moreover, not all middle-aged women were Vitamin D_3_ insufficient at baseline. HMB + D was effective at increasing 25-OH-D on average and achieved similar values as previously published in older adults [35], yet perhaps the benefits are potentiated only in individuals who transition from Vitamin D_3_ insufficiency to sufficiency as shown with IMAT. We were not able to detect any differences in muscle size or function between subjects who were Vitamin D_3_ insufficient versus sufficient at the beginning of the study (data not shown) but our sample size may have limited these exploratory analyses. An alternative hypothesis is that a higher dose of Vitamin D_3_ may be needed to achieve Vitamin D_3_ sufficiency in more participants to realize the full benefits of HMB on skeletal muscle size and function. The current dose of Vitamin D_3_ was half (2000 IU/day) the recommended daily upper limit (4000 IU). Therefore, additional studies could explore the safety and efficacy of a higher Vitamin D_3_ dose with HMB to effectively achieve vitamin D_3_ sufficiency in all participants.

## 5. Conclusions and Limitations

We have completed the first small scale study to suggest that HMB + D supplementation may have some benefits to skeletal muscle and IMAT volume in middle-aged women. It is important to acknowledge that these findings were based on per protocol analysis that excluded 4 non-compliant subjects based on capsule compliance less than 80%. When using the intent to treat analysis that included non-compliant subjects, the differences between placebo and HMB + D for MRI-measured muscle and IMAT volume were no longer different (*p* = 0.1). Therefore, these data may suggest the importance of regular HMB + D intake and that low adherence may diminish any potential benefits. Due to these non-compliant individuals coupled with the impact of COVID-19, a major study limitation is a smaller than intended sample size (*n* = 8–10 per group). Further, the women in this study had a range of Vitamin D_3_ levels at study entry. While this may have helped reveal the potential importance of achieving Vitamin D_3_ sufficiency to lower IMAT with HMB + D, follow up studies should specifically include individuals with Vitamin D_3_ insufficiency to directly test this preliminary association. Additional studies are needed to test whether these findings can be repeated in larger sample size and over a greater time frame to determine if HMB + D could slow or delay the age-related changes in skeletal muscle mass, composition, and function in aging individuals that go from Vitamin D_3_ insufficiency to sufficiency. Since we evaluated the combination of HMB + D, we cannot specifically isolate the impact of either HMB or Vitamin D_3_ on any outcomes. While our data suggest that 12-weeks of HMB + D supplementation may have some benefits on skeletal muscle and IMAT in apparently healthy middle-aged women, additional studies are also warranted to identify if additional populations with high IMAT levels may be prone to benefit from HMB + D.

## Figures and Tables

**Figure 1 nutrients-14-04674-f001:**
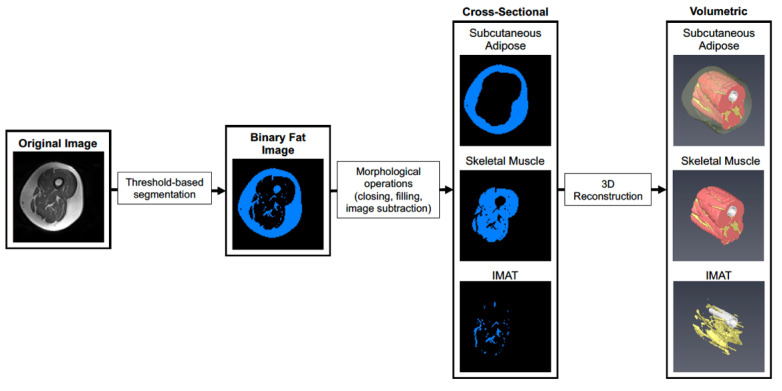
Workflow for segmentation of skeletal muscle and adipose tissue depots. A representative image is shown where the original images are segmented into adipose and skeletal muscle tissue in each cross-section using pixel density and thresholding. Each cross-sectional image is then used to reconstruct a 3D rendering of each tissue across the thigh. The 3D images allow visualization of spatial organization of each tissue and how IMAT infiltrates into skeletal muscle throughout the length of the thigh.

**Figure 2 nutrients-14-04674-f002:**
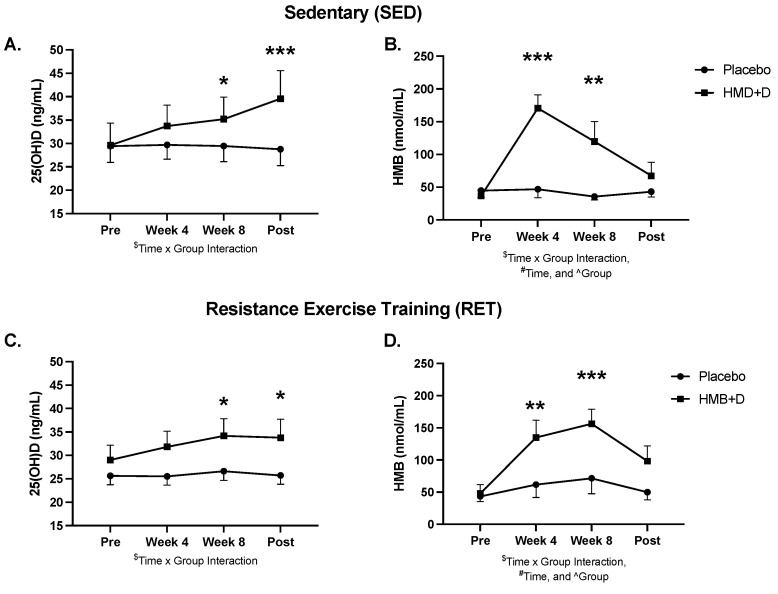
HMB + D increased 25OHD and HMB. Serum 25-OH-D (**A**,**C**) and urine HMB (**B**,**C**) concentration at week 0, 4, 8 and 12 during sedentary control or resistance exercise training. Data analyzed using a two-way ANOVA to detect treatment by time interaction. * *p* < 0.05, ** *p* < 0.01, *** *p* < 0.001 vs. placebo. ^$^ Time x Group interaction; ^#^ main effect for time, and ^ main effect for group. Data presented as mean ± SEM.

**Figure 3 nutrients-14-04674-f003:**
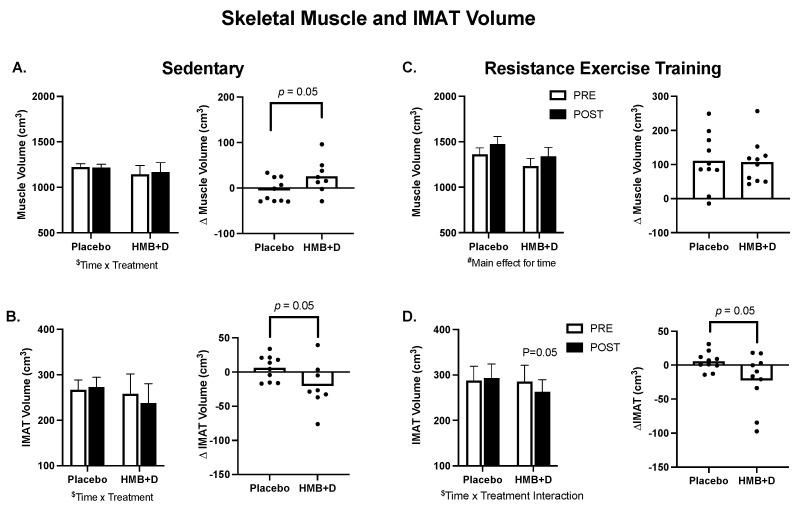
HMB + D decreased thigh IMAT volume during 12 weeks of sedentary behavior or resistance exercise training. Influence of HMB + D on thigh skeletal muscle volume and IMAT during sedentary control (Placebo, *n* = 10; HMB, *n* = 8) (**A**,**B**) and resistance exercise training (Placebo, *n* = 10; HMB + D, *n* = 10) (**C**,**D**). On the left side of each panel, data presented as mean ± SEM. Data analyzed using a two-way ANOVA to detect treatment by time interaction. ^$^ *p* < 0.05 for time × treatment interaction, ^#^
*p* < 0.05 main effect for time. On the right side of each panel, data presented as individual changes (Δ). *p* values are provided.

**Figure 4 nutrients-14-04674-f004:**
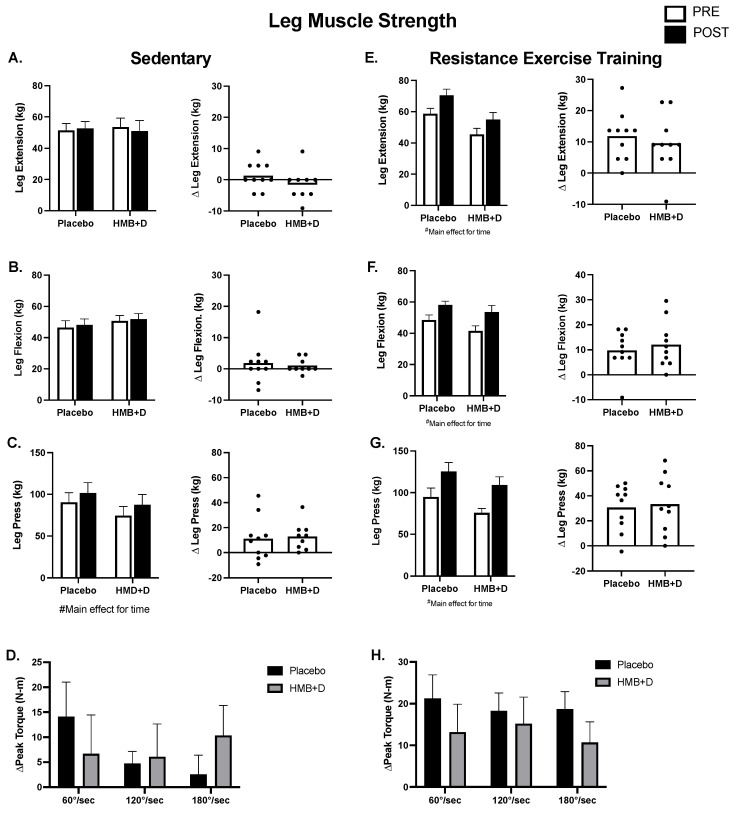
No impact of HMB + D on skeletal muscle function after 12 weeks of sedentary behavior or resistance exercise training. HMB + D did not influence 1-RM leg extension flexion, press nor leg extension isokinetic torque production during sedentary control (Placebo, *n* = 10; HMB, *n* = 9) (**A**–**D**). HMB + D did not augment the increase in 1RM for leg extension flexion, press, or leg extension isokinetic torque production after resistance exercise training (Placebo, *n* = 10; HMB + D, *n* = 10) (**E**–**H**). On the left side of each panel, data presented as mean ± SEM. Data analyzed using a two-way ANOVA to detect treatment by time interaction. ^#^ *p* < 0.05 main effect for time. On the right side of each panel, data presented as individual changes (Δ).

**Figure 5 nutrients-14-04674-f005:**
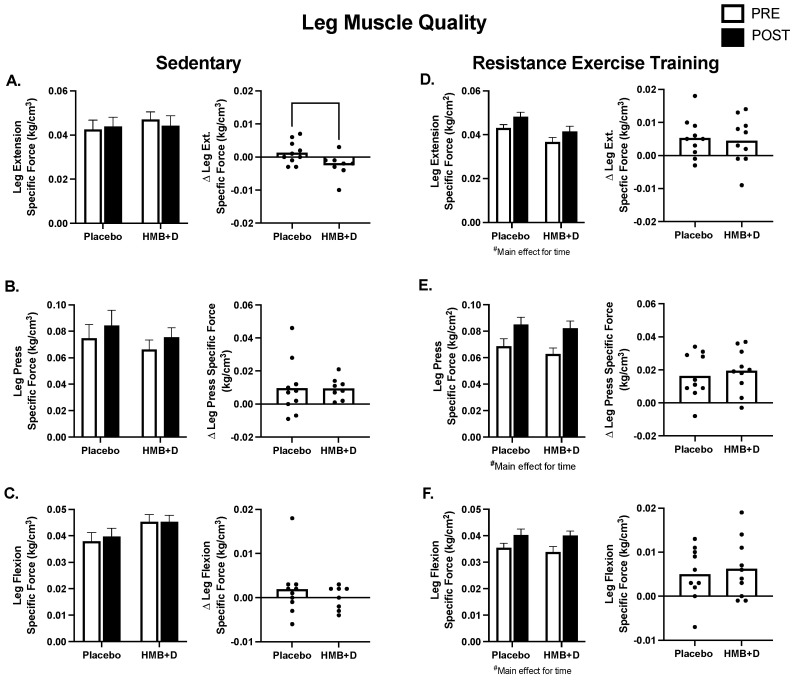
Leg muscle quality after 12 weeks of HMB + D or placebo during sedentary control or resistance exercise training. Leg muscle quality is expressed as specific force (kg per unit of skeletal muscle volume (cm^3^)). Leg extension specific force decreased in HMB + D compared to placebo (**A**). HMB + D did not influence specific force for leg press or leg flexion during sedentary control (Placebo, *n* = 10; HMB, *n* = 8) (**B**,**C**). HMB + D did not influence the increase in specific force for leg extension, leg press, or leg flexion after resistance exercise training (Placebo, *n* = 10; HMB + D, *n* = 10) (**D**–**F**). On the left side of each panel, data presented as mean ± SEM. Data analyzed using a two-way ANOVA to detect treatment by time interaction. ^#^ *p* < 0.05 main effect for time. On the right side of each panel, data presented as individual changes (Δ).

**Figure 6 nutrients-14-04674-f006:**
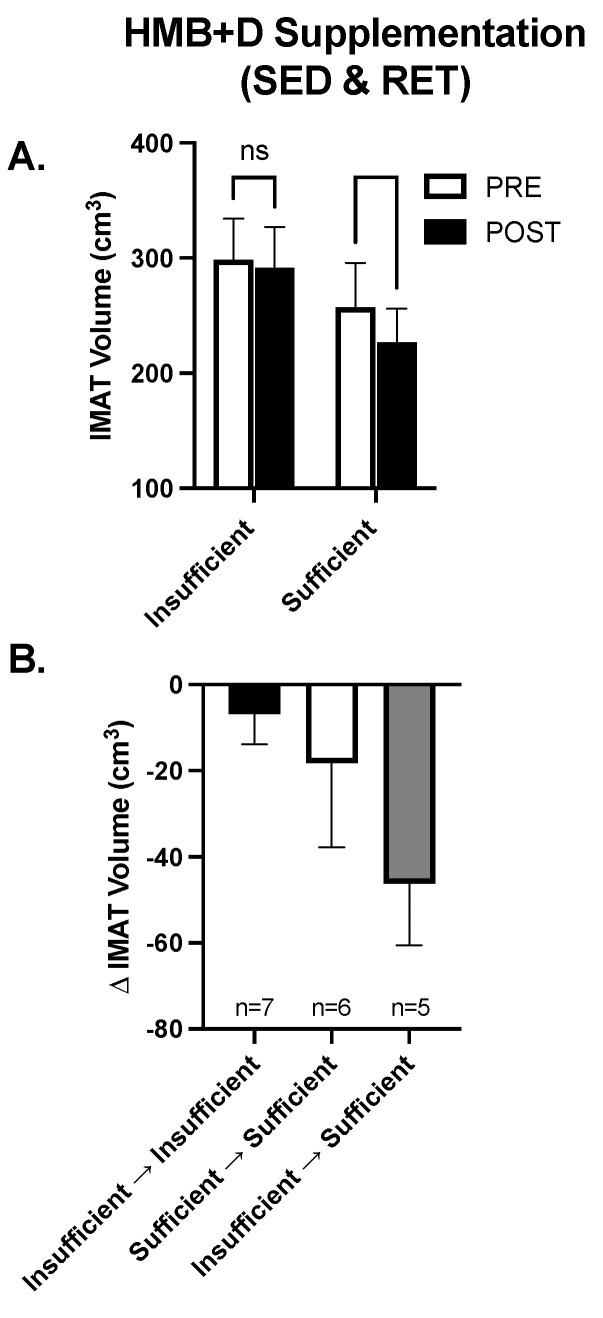
Contribution of Vitamin D_3_ on IMAT after HMB + D supplementation in middle-aged women. Data from women in both the sedentary control and resistance exercise training groups were pooled and then stratified by those who completed the study Vitamin D_3_ insufficient versus sufficient. IMAT was decreased after HMB + D in women who were Vitamin D_3_ sufficient (*n* = 11; SED, *n* = 6; RET, *n* = 5) but not insufficient (*n* = 7, SED, *n* = 3; RET, *n* = 4) (**A**). Women were further stratified into those who remained Vitamin D_3_ insufficient (Insufficient ➔ Insufficient, SED, *n* = 4; RET, *n* = 3), were Vitamin D_3_ sufficient at the start and end of study (Sufficient ➔ Sufficient, SED, *n* = 3, RET, *n* = 4)) and those who transitioned from Vitamin D_3_ insufficient to sufficient (Insufficient ➔ sufficient, SED, *n* = 2, RET, *n* = 3) (**B**). Data presented as mean ± SEM.

**Table 1 nutrients-14-04674-t001:** Self Reported Dietary Intake.

	Sedentary				Resistance Exercise Training			
	Placebo (*n* = 10)		HMB + D (*n* = 9)		Placebo (*n* = 10)		HMB + D (*n* = 10)	
	PRE	POST	PRE	POST	PRE	POST	PRE	POST
**Calories (kcal/day**	1766 ± 462	1856 ± 487	1482 ± 441	1523 ± 363	1581 ± 195	1531 ± 436	1511 ± 549	1728 ± 436
**Protein (g/day)**	74 ± 29	75 ± 19	64 ± 13	64 ± 13	76 ± 32	70 ± 16	71 ± 22	73 ± 20
**Carbohydrate (g/day)**	232 ± 64	229 ± 64	157 ± 60	172 ± 52	169 ± 37	164 ± 79	144 ± 56	191 ± 68
**Fat (g/day)**	60 ± 16	73 ± 25	65 ± 32	66 ± 19	62 ± 16	62 ± 16	66 ± 32	75 ± 31

Mean ± standard deviation.

**Table 2 nutrients-14-04674-t002:** Whole body and regional lean and fat mass.

	Sedentary				Resistance Exercise Training			
	Placebo (*n* = 10)		HMB + D (*n* = 9)		Placebo (*n* = 10)		HMB + D (*n* = 10)	
	PRE	POST	PRE	POST	PRE	POST	PRE	POST
**Age (years)**	53 ± 1		53 ± 1		52 ± 1		51 ± 1	
**BMI (kg/m^2^)**	28 ± 1	28 ± 1	26 ± 2	26 ± 2	27 ± 1	27 ± 1	25 ± 2	25 ± 2
**Total Body Mass (kg)**	73 ± 4	73 ± 4	68 ± 5	68 ± 5	75 ± 4	76 ± 3	68 ± 5	69 ± 5
**Whole Body Lean Mass (kg) ^†^**	41 ± 2	41 ± 2	40 ± 2	40 ± 2	41 ± 2	42 ± 1	38 ± 2	39 ± 2
**Appendicular Lean Mass (kg) ^†^**	17 ± 3	17 ± 3	17 ± 3	16 ± 3	17 ± 0.7	18 ± 0.6	15 ± 0.8	16 ± 0.9
**Sarcopenic Index (kg/m^2^) ^†^**	6.2 ± 0.8	6.2 ± 0.8	6.0 ± 0.7	5.9 ± 0.7	6.3 ± 0.3	6.6 ± 0.3	5.9 ± 0.3	6.3 ± 0.3
**Leg Lean Mass (kg) ^†^**	13.3 ± 0.6	13.5 ± 0.6	13.0 ± 0.8	13.3 ± 0.9	13.6 ± 0.5	14.3 ± 0.5	12.1 ± 0.7	12.8 ± 0.8
**Arm Lean Mass (kg) ^$,†^**	4.0 ± 0.8	3.7 ± 0.8 *	3.7 ± 0.5	3.8 ± 0.6	3.7 ± 0.2	3.9 ± 0.2	3.3 ± 0.2	3.5 ± 0.2
**Body Fat (%)**	45 ± 2	44 ± 2	43 ± 2	42 ± 2	43 ± 1	43 ± 1	40 ± 1	40 ± 1
**Fat Mass (kg)**	32.3 ± 2.5	31.7 ± 2.3	28.5 ± 3.9	28.8 ± 4.0	33.4 ± 2.8	33.7 ± 2.8	29.5 ± 3.4	29.0 ± 3.2
**Trunk Fat Mass (kg)**	15.0 ± 1.5	14.9 ± 1.4	13.8 ± 2.1	14.0 ± 2.3	16.0 ± 1.6	16.0 ± 1.6	13.7 ± 1.8	13.4 ± 1.8
**Leg Fat Mass (kg)**	12.6 ± 0.9	12.4 ± 0.9	10.8 ± 1.5	10.8 ± 1.5	12.4 ± 1.1	12.6 ± 1.2	11.4 ± 1.2	11.5 ± 1.3
**Arm Fat mass (kg)**	3.5 ± 0.3	3.3 ± 0.3	2.8 ± 0.4	2.8 ± 0.4	3.9 ± 0.3	3.9 ± 0.3	3.3 ± 0.4	3.1 ± 0.3

^$^ *p* < 0.05 group × time interaction for SED, * *p* < 0.05 vs. pre, ^†^
*p* < 0.05 main effect for time for RET.

**Table 3 nutrients-14-04674-t003:** Upper Body Muscle Strength.

	Sedentary				Resistance Exercise Training			
	Placebo (*n* = 10)		HMB + D (*n* = 9)		Placebo (*n* = 10)		HMB + D (*n* = 10)	
	PRE	POST	PRE	POST	PRE	POST	PRE	POST
**Chest Press (kg) ^‡,†^**	19.7 ± 1.9	20.9 ± 2.3	18.1 ± 1.5	23.5 ± 3.6 *	20.5 ± 2.3	30.1 ± 3.1	15.9 ± 1.9	24.3 ± 2.9
**Shoulder Press (kg) ^†^**	8.6 ± 1.2	9.9 ± 1.3	9.6 ± 1.3	10.1 ± 1.4	9.0 ± 1.0	13.5 ± 1.3	8.2 ± 1.0	13.9 ± 1.8
**Seated Row (kg) ^†^**	30.0 ± 2.0	30.0 ± 2.1	29.2 ± 1.5	28.5 ± 1.1	30.2 ± 1.7	38.2 ± 2.0	25.8 ± 1.9	34.5 ± 2.6

* *p* < 0.05 vs. pre, ^‡^ *p* < 0.05 main effect for time for SED. ^†^ *p* < 0.05 main effect for time for RET.

## Data Availability

The data presented in this study may be available on request from the corresponding author and appropriate data transfer and use agreements.

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
