# Peer review of "Small-Scale Randomized Controlled Trial to Explore the Impact of β-Hydroxy-β-Methylbutyrate Plus Vitamin D3 on Skeletal Muscle Health in Middle Aged Women"

_nutrients, 2022, doi:10.3390/nu14214674_

Round 1

Reviewer 1 Report

This randomized controlled trial indicated that the combination of HMB with vitamin D3 significantly increased muscle volume in middle-aged women in the sedentary group while decreased IMAT in those in both the sedentary and exercise group compared with their counterparts receiving placebo. I have some concerns with this study regarding the methodology and reporting, although this research handled the exciting issue.

Major comments:

#1 As the authors noticed, the major limitation of the study was the small sample size. The number of participants may not be sufficient at the design phase(n=48), but it is not sure because no justification of sample size was provided. Please justify the statistical sufficiency of the original sample size based on the primary outcome (see also #2). Even if the sample size were established, it would be critical to determine and discuss the effect of decreasing the sample size to 43 patients in the Discussion session.

#2 Please specify the study hypothesis in detail. What were the primary and secondary outcomes? These issues should be along with the study protocol.

#3 The investigators should describe the type of randomization (e.g., unrestricted randomization, block randomization, etc) and how they randomized the patients (e.g., using a random table, closed envelope, etc.).

#4 Whether the allocation was concealed for the investigators was not described. It is essential if the investigators with COI with the manufacturer of HMB (JAR and LMP) are concerned with randomization regardless of allocation concealment. Please describe the details.

#5 The authors stated that the study employed double-blinding fashion, but it is unclear how they did the blinding procedure. Notably, it is vital to describe whether the person who collected the baseline data and outcome measures, and those who analyzed the data were blinded throughout the study period.

#6 The way of analysis (per protocol or ITT) was not described in the Statistical Analysis section. Was the per-protocol analysis truly established before conducting the study? If so, why the investigators chose it? The authors should state the analytical method based on the original protocol.

#7 Results: The numbers and characteristics of the participants in each group should be described at the start of the Results section. The authors should extract the baseline characteristics from Table2 and make a new table (Table 1). Between-group comparisons for all baseline characteristics must be made to identify the appropriateness of the randomization.

#8 I could not determine the correctness of the results before confirming the pre-established analytical strategy, including the justification of sample size, the reason for per-protocol analysis, and the plausibility of other analytical methods. In my opinion, too many statistical analyses were performed in the current study without setting the primary outcome, indicating an increase in the probability of obtaining statistically significant results by chance.

#9 The sub-group analyses for the effects of HMB+VD on IMAC per serum VD concentrations (Fig.6) were misleading. Three sub-groups (Insufficient-insufficient, sufficient-sufficient, insufficient-sufficient) may involve both SED and RET arms, so the results should be interpreted by taking the arm allocated into account. I suggest these results should be removed.

#10 The same as #8, the discussion section could not be assessed before addressing the methodological concerns. The style of writing in the discussion seems too assertive based on the results from a small sample.

Author Response

We appreciate the feedback regarding our study design and statistical considerations. We have made our best effort to address the reviewers concern and will make appropriate changes to improve the manuscript.

Major comments:

1 As the authors noticed, the major limitation of the study was the small sample size. The number of participants may not be sufficient at the design phase(n=48), but it is not sure because no justification of sample size was provided. Please justify the statistical sufficiency of the original sample size based on the primary outcome (see also #2). Even if the sample size were established, it would be critical to determine and discuss the effect of decreasing the sample size to 43 patients in the Discussion session.

We acknowledge the small sample size, however, our original target was supported by power calculations from PMID: 32857128. Based on the reviewers feedback, we have included our sample size justification within the manuscript and below. In our hands performing similar studies, we have never had issues with participant retention and we could have never foreseen the impact COVID-19 would have on subject enrollment, retention and recruitment. We sincerely hope the reviewer understands the difficult nature of completing human subjects research during COVID-19 while on a finite budget and timeline.

“Since no studies using HMB+D have been previously performed in middle-aged women, power analysis (G-Power, Universität Kiel, Germany) was completed based on lean body mass and knee extensor function from a previous study by Ratchmacher et al. testing HMB+D in older adults. For the power analysis, we used a previously measured 0.44 kg change in lean body mass with HMB+D and a -0.37 Kg change in lean body mass for placebo to support our primary outcome of thigh skeletal muscle volume. Based on a F-Test (ANOVA) with repeated measure for time (pre vs. post) and 4 groups with a power of 0.8 and an  α-error probability of 0.05, we estimate the need for a total of 44 subjects. Using the same power analysis approach, we also used a change in knee extensor function of 10.9 Nm for HMB+D and -5.2 Nm for control to support our secondary endpoint of knee extensor function. We determined a need for a total of 48 participants (n=12/group) for indices of knee extensor function.

An a priori 80% compliance rate was set as this permitted subjects to miss up to 1 day of capsules per week and was lower than our average compliance rates in previous studies using nutritional supplementation or commonly consumed medications (³90%). Since the goal of this study was to determine the impact of HMB+D in middle-aged women randomized to sedentary or resistance exercise training groups, all statistical analyses were completed within each group.

2 Please specify the study hypothesis in detail. What were the primary and secondary outcomes? These issues should be along with the study protocol.

The primary outcome of this study was skeletal muscle volume. The secondary outcomes was knee extensor function. Exploratory outcomes were intermuscular adipose tissue (IMAT), muscle quality (knee extensor function per unit muscle volume), and the potential influence of VitD status on skeletal muscle size, function and IMAT. 

This study tested the hypothesis that 12-weeks of HMB+D supplementation would 1) increase skeletal muscle size, function, composition and quality in sedentary middle-aged women and 2) augment the increase in skeletal muscle size, function, composition and quality in middle-aged women completing resistance exercise training. 

3 The investigators should describe the type of randomization (e.g., unrestricted randomization, block randomization, etc) and how they randomized the patients (e.g., using a random table, closed envelope, etc.).

After participants were determined to be eligible, they were randomized to one of four groups in random block sizes of 4 or 8 to minimize selection bias. The randomization sequence was created in STATA. 

4 Whether the allocation was concealed for the investigators was not described. It is essential if the investigators with COI with the manufacturer of HMB (JAR and LMP) are concerned with randomization regardless of allocation concealment. Please describe the details.

The entire author list remained blinded until the after data analysis of muscle and IMAT, knee extensor function, body composition, and all blood/urine samples. The randomization was performed by the investigative team at UIUC/UW-Madison. JAR and LMP were not involved in randomization. JAR and LMP sent a sealed envelope that contained which capsules were placebo vs. supplement. The envelope was opened upon completion of data analysis outlined above. 

5 The authors stated that the study employed double-blinding fashion, but it is unclear how they did the blinding procedure. Notably, it is vital to describe whether the person who collected the baseline data and outcome measures, and those who analyzed the data were blinded throughout the study period.

Similar to #4, the study used matching capsules of placebo and supplement. The study team (data collection, data analysis) remained blinded to placebo vs. supplement until completion of data analysis. Serum and urine measurements of 25-OH-D and HMB were performed at the end of the study as this would have reveled who was taking the supplement.

6 The way of analysis (per protocol or ITT) was not described in the Statistical Analysis section. Was the per-protocol analysis truly established before conducting the study? If so, why the investigators chose it? The authors should state the analytical method based on the original protocol.

The a priori compliance cutoff was set at 80% based on our previous studies using dietary supplements and medications that averaged ≥90%. Further, 80% compliance permitted participants to miss pills no more than 1 day per week. A priori, our focus was on the physiological effects of receiving the assigned treatment (placebo vs. supplement) and therefore we chose per protocol analysis. Additionally, we transparently report that when performing intention to treat analysis that the statistical difference becomes non-signficant (P=0.1). 

7 Results: The numbers and characteristics of the participants in each group should be described at the start of the Results section. The authors should extract the baseline characteristics from Table2 and make a new table (Table 1). Between-group comparisons for all baseline characteristics must be made to identify the appropriateness of the randomization.

As requested will include all baseline characteristics in a new Table and test whether there were any baseline differences between subjects. Importantly, subjects were randomized to four groups, however, we did not stratify to match physical characteristics so it would not be surprising if these characteristics were not identical. 

8 I could not determine the correctness of the results before confirming the pre-established analytical strategy, including the justification of sample size, the reason for per-protocol analysis, and the plausibility of other analytical methods. In my opinion, too many statistical analyses were performed in the current study without setting the primary outcome, indicating an increase in the probability of obtaining statistically significant results by chance.

The reviewers concerns will be addressed within the manuscript. We used two-way ANOVA to test hypothesis 1 and hypothesis 2. We also present individual data for the delta between pre and post for readers to appreciate the inter-subject variability. These analyses were consistent with our original analysis (2-Way ANOVA).

9 The sub-group analyses for the effects of HMB+VD on IMAC per serum VD concentrations (Fig.6) were misleading. Three sub-groups (Insufficient-insufficient, sufficient-sufficient, insufficient-sufficient) may involve both SED and RET arms, so the results should be interpreted by taking the arm allocated into account. I suggest these results should be removed.

We agree that these data are exploratory and we articulate the preliminary nature of these exciting associations in the manuscript. However, we think these data would be of interest to the research community and readership and may help spur future scientific inquiry to directly test these associations in future trials. As recommended by the reviewer, we will detail how many participants are included in each group and underscore the exploratory nature of these data.

10 The same as #8, the discussion section could not be assessed before addressing the methodological concerns. The style of writing in the discussion seems too assertive based on the results from a small sample.

Throughout the discussion we discuss our observations as "may",  "suggest", or "imply." We feel these statements appropriately place our observational findings in the appropriate context of a small-scale trial and do not over interpret our findings.

Reviewer 2 Report

The article titled "Small-Scale Randomized Controlled Trial to Explore the Impact of β-Hydroxy-β-methylbutyrate plus Vitamin D3 on Skeletal Muscle Health in Middle Aged Women" is very interesting. The purpose of this study was to determine if supplementation with calcium HMB+D, aimed at achieving sufficient circulating levels of 25-OH-D, could improve skeletal size, function, composition, and quality in middle-aged women (45-60 yrs) during 12-weeks of a non-exercise sedentary control (SED) or resistance exercise training (RET) program. The study is well structured, the limitations have been described by the authors.

I would like to ask the authors the following questions:

how was the sample size calculated?

is it possible to insert the study flow-chart?

image A in Figure 3 is missing

Is it possible to insert in the tables and figures the number of subjects enrolled in each group?

Author Response

Reviewer 2: We would like to thank the reviewer for their important clarifying questions.

The article titled "Small-Scale Randomized Controlled Trial to Explore the Impact of β-Hydroxy-β-methylbutyrate plus Vitamin D3 on Skeletal Muscle Health in Middle Aged Women" is very interesting. The purpose of this study was to determine if supplementation with calcium HMB+D, aimed at achieving sufficient circulating levels of 25-OH-D, could improve skeletal size, function, composition, and quality in middle-aged women (45-60 yrs) during 12-weeks of a non-exercise sedentary control (SED) or resistance exercise training (RET) program. The study is well structured, the limitations have been described by the authors.

I would like to ask the authors the following questions:

how was the sample size calculated?

Since no studies using HMB+D have been previously performed in middle-aged women, power analysis (G-Power, Universität Kiel, Germany) was completed based on lean body mass and knee extensor function from a previous study by Ratchmacher et al. testing HMB+D in older adults. For the power analysis, we used a previously measured 0.44 kg change in lean body mass with HMB+D and a -0.37 Kg change in lean body mass for placebo to support our primary outcome of thigh skeletal muscle volume. Based on a F-Test (ANOVA) with repeated measure for time (pre vs. post) and 4 groups with a power of 0.8 and an  α-error probability of 0.05, we estimate the need for a total of 44 subjects. Using the same power analysis approach, we also used a change in knee extensor function of 10.9 Nm for HMB+D and -5.2 Nm for control to support our secondary endpoint of knee extensor function. We determined a need for a total of 48 participants (n=12/group) for indices of knee extensor function.

is it possible to insert the study flow-chart?

Based on the small scale nature of this study we did not originally include. We can insert the study flow-chart into the supplemental files.

image A in Figure 3 is missing

We will correct figure 3 to begin with Figure 3A.

Is it possible to insert in the tables and figures the number of subjects enrolled in each group?

We will include subject numbers in the tables and figures.

Round 2

Reviewer 1 Report

The authors made a revision that seems appropriate in some parts, but others remained.

#1 Small sample size due to Covid-19 pandemic was understandable, but it does not justify the suboptimal scientific validity.

#2 The primary and secondary outcome remained unclear. What did "skeletal muscle volume" and "knee extensor muslce function" stand for? The sample size caluculation that used whole body LBM, guessing the whole body lean mass was the primary outcome. However, this outcome did not different between the groups. These vague descriptions of the outcomes significantly hamper the reliability of this study.

#3 The differences of the charactaristics of the individuals between the groups suggest that the randomization was insufficient, indicating biased results. 

Author Response

1 Small sample size due to Covid-19 pandemic was understandable, but it does not justify the suboptimal scientific validity.

We appreciate the reviewers concern regarding small sample size. We believe we have adequately addressed this concern in the manuscript and in the previous review cycle. Unfortunately, the reviewer does not provide any question, concern, or feedback that we can respond to further improve the manuscript. We thank the reviewer for their valuable time and enthusiasm for our study. 

#2 The primary and secondary outcome remained unclear. What did "skeletal muscle volume" and "knee extensor muslce function" stand for? The sample size caluculation that used whole body LBM, guessing the whole body lean mass was the primary outcome. However, this outcome did not different between the groups. These vague descriptions of the outcomes significantly hamper the reliability of this study.

We apologize for any confusion. Our primary outcome was thigh skeletal muscle volume assessed by MRI. We define our region of interest and the approach to measure thigh skeletal muscle volume in the methods section. We chose whole body lean mass to calculate sample size because this was the only available muscle volume/mass related outcome using HMB+D. We and others have previously found that gold-standard MRI is a more sensitive approach to detect changes in thigh skeletal muscle cross-sectional area and volume than DEXA lean mass (PMID: 19692660, 20566734, 22984247). Therefore, we are confident that our a priori sample size based on DEXA derived whole-body lean mass was appropriately powered to detect any differences between groups in our primary outcome of MRI measured thigh skeletal muscle volume. 

3 The differences of the charactaristics of the individuals between the groups suggest that the randomization was insufficient, indicating biased results.

As noted in the revised manuscript (ln 356-357), there were no statistical differences between groups for baseline characteristics. Samples were randomized as described in the methods section (and pasted below). We believe that by using these standard approaches, randomization would minimize selection bias. 

"Subjects were randomized to one of four groups which included placebo or HMB+D supplementation during 12 weeks of non-exercise sedentary control (SED) or a progressive resistance exercise training program (RET). The randomization sequence was created in STATA and was performed in random block sizes of 4 or 8 to minimize selection bias. The randomization was performed by the investigative team at UIUC/UW-Madison and was concealed to all study team members, including those employed by MTI. Further, the study blind was maintained by the study team until after data analysis."